# Magnetic TiO_2_/CoFe_2_O_4_ Photocatalysts for Degradation of Organic Dyes and Pharmaceuticals without Oxidants

**DOI:** 10.3390/nano12193290

**Published:** 2022-09-21

**Authors:** Islam Ibrahim, George V. Belessiotis, Ahmed Mourtada Elseman, Mohamed Mokhtar Mohamed, Yatao Ren, Tarek M. Salama, Mahmoud Basseem I. Mohamed

**Affiliations:** 1Department of Chemistry, Faculty of Science, Al-Azhar University, Cairo 11884, Egypt; 2Chemical Engineering Department, National Technical University of Athens NTUA, 15780 Athens, Greece; 3Electronic and Magnetic Materials Department, Advanced Materials Institute, Central Metallurgical Research and Development Institute, Cairo 11421, Egypt; 4Egypt-Japan University of Science and Technology, Borg El Arab, Alexandria 21934, Egypt; 5Chemistry Department, Faculty of Science, Benha University, Benha, Egypt; 6Harbin Institute of Technology, School of Energy Science and Engineering, Harbin, China; 7Faculty of Engineering, University of Nottingham, University Park, Nottingham, UK

**Keywords:** TiO_2_ composite, visible light, tetracycline, photocatalytic oxidation, magnetic composite

## Abstract

In the current study, CoFe_2_O_4_ and TiO_2_ nanoparticles were primarily made using the sol-gel method, and subsequently, the hybrid magnetic composites of TiO_2_ loaded with CoFe_2_O_4_ (5–15 percent *w*/*w*) were made using a hydrothermal procedure. X-ray diffraction (XRD), Fourier transform infrared (FTIR) and Raman spectroscopy, ultraviolet-visible diffuse reflectance spectroscopy (UV-vis DRS), scanning electron microscopy (SEM), and high-resolution transmission electron microscopy (HRTEM) were all used to thoroughly characterize the materials. Additionally, the zero-charge point (ZCP) determination, the examination of the pore structure by nitrogen adsorption, and an evaluation of magnetic properties were performed. Six organic dye pollutants were selected to evaluate the performance of the synthesized nanocomposites toward photocatalytic degradation, including methylene blue (MB), methyl orange (MO), crystal violet (CV), acridine orange (AO), rhodamine B (RhB), and rhodamine 6G (R-6G). Photodegradation of tetracycline (TL), a model pharmaceutical pollutant, was also studied under UV and visible light. The composites exhibited a high degradation performance in all cases without using any oxidants. The photocatalytic degradation of tetracycline revealed that the CoFe_2_O_4_/TiO_2_ (5% *w*/*w*) composite exhibited a higher photocatalytic activity than either pure TiO_2_ or CoFe_2_O_4_, and thus attained 75.31% and 50.4% degradation efficiency under UV and visible light, respectively. Trapping experiments were conducted to investigate the photodegradation mechanism, which revealed that holes and super oxide radicals were the most active species in the photodegradation process. Finally, due to the inherent magnetic attributes of the composites, their easy removal from the treated solution via a simple magnet became possible.

## 1. Introduction

As a result of the use of dyes for coloring by the textile, plastics, and other industries, the amount of organic dye pollution has increased [1]. Industrial dyes are one of the most prominent water pollutants [2]. Some of the most infamous of these organic dye pollutants (ODP) include: (a) Methylene Blue (MB), known for its toxicity, carcinogenic nature, and non-biodegradability [2], (b) Crystal Violet (CV), which can cause issues from permanent eye injuries to respiratory or kidney failure [3], (c) Methyl Orange (MO), an azo dye that is mutagenic and carcinogenic and has a limited biodegradability [4], (d) Acridine Orange (AO), a cell-permeable toxic dye with mutagenic potential [5], (e) Rhodamine B (RhB), a dye which poses a threat for the respiratory tract and skin with carcinogenic and neurotoxic potential [6] and (f) Rhodamine 6G (R-6G), another dye of the rhodamine family, which presents similar dangers [7]. The chemical structures of the selected six ODP are shown in Figure 1. Pharmaceutical ingredients are another major source of pollutants. They are considered a significant threat to humans’ endocrine, reproductive, and cognitive systems [8]. One such pharmaceutical pollutants is the antibiotic Tetracycline (TL), which often finds its way into the environment, and can accumulate in water due to its high hydrophilicity and cause numerous problems in ecosystems and humans [9]. While numerous methods have been developed to deal with ODP, many of these methods require extra processes, since byproducts are frequently present.

Photocatalysis is an affordable green water treatment technology, which has the potential to be very efficient in the decomposition of the ODP [10,11], as the photo-oxidation of dyes, for example, can result in their mineralization into simpler species without toxicity [2]. Especially prevalent is the use of composite photocatalytic materials [12,13]. With a conscious choice of individual materials, a composite can be comprised of components that complement each other, leading to enhanced performance and utility. The most well-known photocatalyst is titanium dioxide (TiO_2_) [14,15], especially its anatase phase, due to its high stability and outstanding photocatalytic performance under ultraviolet (UV) light. However, it has significant drawbacks regarding its wide band gap, which renders it photocatalytically active only under UV light, and its fast charge pair recombination impairs its photocatalytic performance. Thus, the formation of composites with TiO_2_ evolved into a common strategy, since it can rectify its weaknesses [1]. The formation of such a photocatalytic heterojunction can enhance factors such as charge separation, thereby enhancing photocatalytic performance [15,16]. Magnetic materials, on the other hand, have several interesting properties [17,18,19], and their combination with TiO_2_ is promising [20,21], as the added magnetic properties offer increased utility for the composite: In magnetic composites, after the photocatalytic treatment of a solution is completed, the photocatalyst can be easily removed with a simple magnet, which facilitates its re-usability [22,23]. Spinel ferrites are a prominent family of magnetic ferrite materials with an AFe_2_O_4_ structure (A = divalent cation), characterized by a high saturation magnetization, among other favorable features [24]. Thus, the combination of TiO_2_ with a spinel ferrite material appears especially promising. This work presents a complete overview of the TiO_2_/spinel ferrite nanocomposites with an extensive evaluation of their photocatalytic capabilities against several of the more well-known organic pollutants. It also presents the potential of the nanocomposites for recyclable usage, taking advantage of their magnetic retrieval ability, thereby highlighting the extensive capacities of the TiO_2_/spinel ferrite nanocomposites.

In this work, a hydrothermal process was employed to synthesize composites comprised of the CoFe_2_O_4_-loaded TiO_2_. To optimize the composite compositions, three weight percentage ratios of CoFe_2_O_4_ to TiO_2_ corresponding to 5, 10, and 15% *w*/*w* were synthesized. A complete characterization of the composites against the starting materials was undertaken via XRD, FTIR, Raman, UV–vis DRS, SEM, and HR-TEM. The structure analysis verified the successful formation of the CoFe_2_O_4_/TiO_2_ hybrid nanocomposites. The photocatalytic performances of the hybrid composites were evaluated in the degradation of six different ODP under UV light and the degradation of tetracycline (TL) under UV and visible light.

## 2. Experimental Methods

### 2.1. Materials

The materials used in this research work are presented below. From Acros Organics: Absolute ethanol (≥99%), ferric nitrate (≥99%), titanium (IV), n-butoxide (≥99%), benzoquinone, oxalic acid (≥98%), and tetracycline (≥99%). From Sigma-Aldrich: glacial acetic acid (≥98%). From Chem-Lab: cobalt nitrate (≥99%), HCl. From Sigma-Aldrich: Methylene Blue, Methyl Orange, Crystal Violet, Acridine Orange, Rhodamine B, and Rhodamine 6G were purchased.

### 2.2. Material Preparation

#### 2.2.1. Synthesis of TiO_2_ Nanoparticles

The sol-gel method was utilized for the preparation of titanium dioxide nanopowder [25,26]. In a typical process, a solution of 10 mL of Ti (IV) n-butoxide and 40 mL absolute ethanol (A) and a solution of 4 mL DI H_2_O, 2 mL acetic acid, 10 mL absolute ethanol along with HCl (10% *wt*/*wt*) for pH adjustment to ~2 (B) were mixed by the dropwise addition of (B) to (A) and stirring (1 h). A yellow translucent gel was formed 3 h after the end of the stirring and after drying overnight at 100 °C; yellow crystallites were formed, which produced a white powder after grinding. Finally, this powder was calcinated (300 °C for 1 h and 450 °C for 2 h), producing our nano-TiO_2_ sample (T).

#### 2.2.2. Synthesis of CoFe_2_O_4_

The cobalt ferrite was also prepared via a sol-gel technique [11]. A total of 1.9 g of oxalic acid (complexing agent) was slowly added to a mixture of cobalt nitrate (0.01 moles) and ferric nitrate (0.02 moles), under stirring (1 h). After evaporation (80 °C under stirring), the samples were dried (110 °C, 24 h) and annealed (1000 °C, 2 h).

#### 2.2.3. Synthesis of CoFe_2_O_4_/TiO_2_ Composites

For the preparation of the final composites, a hydrothermal method was adopted [11]. Samples with three different *w*/*w* % concentrations of CoFe_2_O_4_ to TiO_2_ were prepared by varying the initial quantity of CoFe_2_O_4_ in the process: 0.05 (5% *w*/*w*), 0.1 (10% *w*/*w*), and 0.15 (15% *w*/*w*) g of CoFe_2_O_4_ was ultrasonically treated in 100 mL of deionized water (2 h) before the adding of 0.95, 0.9, and 0.85 g of TiO_2_, respectively. A total of 10 g of urea was also added as a combustion fuel. After further ultrasonication (1 h), each solution was put in a 300 mL stainless steel autoclave coated with Teflon, where they were heated (200 °C, 12 h) before the natural cooling of the autoclave (room temperature). After washing and drying, we acquired the TC5, TC10, and TC15 samples, each with an index signifying its % *w*/*w* of CoFe_2_O_4_ (C) and TiO_2_ (T).

### 2.3. Characterization of Samples

For the characterization of the prepared photocatalysts, we utilized X-ray diffraction (XRD, D8-ADVANCE), FTIR (Perkin Elmer Spectrum 100 Spectrometer) and micro Raman (Jobin-Yvon LabRam) vibrational spectroscopy, SEM (Jeol JSM 7401F FieldEmission), HR-TEM (high-resolution JEOL JEM-2100 LaB6), UV-Vis (Hitachi U-4100) spectroscopy, Magnetic measurements were performed via a VSM (9600-1 LDJ, Lake Shore, Columbus, OH, USA) instrument. Textural characterization of the samples was carried out by the N_2_ adsorption-desorption at −196 °C. Prior the measurements, the samples were outgassed at 200 °C for 48 h. The Brunauer–Emmett–Teller (BET) equation was applied to determine the specific surface area (S_BET_). The BJH method was applied to the desorption branch of the N_2_ isotherms to obtain the meso-pore volume (V_meso_) and cumulative surface area (S_BJH_) of the mesopores. A batch technique was utilized for the determination of zero charge point (ZCP) using six beakers, each containing 0.1 M KCl (12.5 mL) with an assumed initial pH value of 4–12 using 0.1 M HCl or KOH [27]. After adding a standard quantity of a photocatalyst to each beaker, stirring for 24 h until the pH was at equilibrium, the sample was removed by filtration. The ZCP values were determined by graphing the final pH of the solution against the initial pH.

A total of 5 mg of the photocatalyst was added to an ODP or TL solution (50 mL, 10 ppm) to evaluate its photocatalytic degradation performance. After stirring for 1 h in dark conditions (adsorption-desorption equilibrium), the solution was illuminated (Osram L BLUE UVA 15 W/78 lamps), and 4-daylight Sylvania,—F15W/53-765-T8 (450–710 nm) lamps in a photoreactor box. The progress of the photocatalytic degradation was evaluated at standard time intervals by measuring the absorption peak of each dye solution through UV-Visible spectroscopy. The characteristic absorption peaks for the six ODP are: MB (664 nm), CV (585 nm), MO (464 nm), AO (490 nm), RhB (553 nm), Rh6G (526 nm), and (357 nm) for the TL solution. Finally, to study the active species responsible for the photocatalytic degradation, scavenger tests were performed by repeating the above process in the presence of a scavenger (for each studied species) in the solution: KI (h^+^ quencher), BQ (O_2_^−^ quencher), KBrO_3_ (e^−^ quencher), and IPA (OH^.^ quencher) [14].

## 3. Results and Discussion

### 3.1. Structural Analysis

In terms of a crystal structure analysis, all samples were analyzed with XRD and the resulting patterns are shown in Figure 2. For TiO_2_, the expected diffraction peaks and the corresponding Miller indices appear at 2θ of 25.8° (101), 36.9° (103), 37.8° (004), 38.5° (112), 48.0° (200), 53.8° (105), 55.0° (211), 62.8° (204), 68.7° (116), and 70.3° (220), verifying the presence of the pure anatase TiO_2_ (I4_1_/amd (141) space group, JCPDS Card number 21-1272] [28]. For CoFe_2_O_4_, the peaks at 2θ of 18.2° (111), 30.0° (220), 35.4° (311), 37.0° (222), 43.0° (400), 53.4° (422), 56.9° (511) and 62.5° (440) correspond to spinel ferrite with a cubic symmetry (Fd-3m (227) space group, and the JCPDS Card no. 22-1086) [24] were depicted. The diffraction patterns for the TC5, TC10, and TC15 hybrid nanocomposites present peaks that correspond to both starting CoFe_2_O_4_ and TiO_2_ materials, signifying the good chemical hybridization of the two oxides. Following the insertion of CoFe_2_O_4_, a considerable decrease in the (101) anatase peak intensities was measured, demonstrating a strong interaction between the components constituting the composites, notably the TC10 sample.

Vibrational spectroscopy, via Raman [29,30] and FT-IR measurements [31], was also utilized to study the samples’ constitution and crystallinity. The corresponding analyses are presented in Figure 3.

Concerning Raman spectroscopy, (Figure 3a): For the TiO_2_ sample, we observed the Raman active modes at 141 cm^−1^ (E_g_), 196 cm^−1^ (E_g_), 394 cm^−1^ (B_1g_), 513 cm^−1^ (A_1g_ + B_1g_), and, lastly, 640 cm^−1^ (E_g_) [32], verifying the anatase-TiO_2_ crystallization. For the CoFe_2_O_4_ sample, its characteristic bands appear at 175 cm^−1^ (T_2g_), 295 cm^−1^ (E_g_), 468 cm^−1^ (T_2g_), and 613 cm^−1^ (T_2g_) [33]. As for the composite materials, the characteristic peaks of the individual components appear in the final materials, proposing the amalgamation of the two components forming the composites. In the FT-IR spectra (Figure 3b), TiO_2_ exhibits a broad absorption band (~469–840 cm^−1^) related to Ti-O-Ti vibration bonds [34], while CoFe_2_O_4_ exhibits a peak at ~469 cm^−1^ (stretching vibration in Fe(III)-O^2−^) and another peak at ~586 cm^−1^ (stretching vibration in Co(II)-O^2−^) [35]. Thus, a broad absorption at ~462–580 cm^−1^ appears in the final composites. Thus, the proper merging of TiO_2_ and CoFe_2_O_4_ is verified from either type of vibrational spectroscopy analysis.

### 3.2. Morphological Analysis

SEM and TEM microscopy were employed to assess the morphology of the synthesized samples concerning the starting components (Figure 4). While TiO_2_ nanoparticles (NPs) appear distinct, there is an agglomeration of CoFe_2_O_4_ nanoparticles, since there are magnetic forces between them [21], making CoFe_2_O_4_ nanostructures significantly larger than TiO_2_ NPs (TiO_2_ NPs are under 20 nm, while CoFe_2_O_4_ clusters are over 120 nm in size). The SEM picture of TC10 (Figure 4c), which depicts the final composites, demonstrates how the CoFe_2_O_4_ nanostructures’ surfaces have been suitably ornamented with TiO_2_ NPs. As observed by the TEM picture (Figure 4d), titania has actually been dispersed over the grains of the ferrite material.

### 3.3. Surface Analysis

N_2_ adsorption-desorption isotherms and pore-size distribution plots for each sample have been shown in Figure 5 and Figure 6 to determine the samples’ surface area and porosity.

A Type IV isotherm corresponding hysteresis loop is presented by the base-TiO_2_ sample (Figure 5a), signifying mesoporous materials [36], while there is no hysteresis loop present for CoFe_2_O_4_ (Figure 5b), signifying the type III isotherm [37]. The type IV isotherm is also representative of the composite materials (Figure 5c), meaning they are mesoporous. The specific surface values (m^2^/g) for the samples were: TiO_2_ (40), CoFe_2_O_4_ (3.2), TC5 (39.1), TC10 (38.2), and TC15 (37). There was a slight decrease in the BET surface area value when going from TiO_2_ to the TiO_2_/CoFe_2_O_4_ composites due to the low value in the case of the ferrite. Pore size distribution reveals no significant differences for the unimodal type of pore presented, either from the mother TiO_2_ or the composite TC10, with a pore maximized at around 52 Ă compared to the bimodal type of pores depicted for CoFe_2_O_4_ at 30 and 40 Ă. This demonstrates how the composites maintain their mesoporosity sequence. Regarding the surface chemistry of the samples, the zero charge point (ZCP) evaluation (Figure 7) took place for samples TiO_2_ (4.45), CoFe_2_O_4_ (6.95) and TCX (x-5,10,15) (4.9) samples. At solution pH > ZCP pH, the photocatalyst can favor the adsorption of positively charged pollutants such as the positively charged MB [38], (with negatively charged contaminants being favored in the opposite case) [39]. Thus, at a higher solution pH (negatively charged surface), there can be increased pollutant adsorption of dyes such as MB [40] onto the TCX surface.

### 3.4. Optical Analysis

To learn more about the photocatalysts’ optical characteristics, UV-vis spectroscopy was employed (Figure 8). The band gaps for TiO_2_ (3.1 eV), CoFe_2_O_4_ (1.37 eV), TC5 (3.18 eV), TC10 (3.16 eV), TC15 (3.07 eV) were calculated via the Kubelka-Munk equation and the Tauc plots (derived from diffuse reflectance spectra) (Figure 8a). As for the absorption spectra (Figure 8b), the absorption edge for TiO_2_ is located near 400 nm, as expected [41], while the spinel ferrite absorbs significantly in the visible region [37]. As a result, the composite materials have enhanced absorption in the visible range.

### 3.5. Magnetization Analysis

The value of saturation magnetization (M_s_) is important for a photocatalyst as it signifies its capability for magnetic removal from a treated solution. The magnetization curves of all magnetic samples are presented in Figure 9.

As expected, the composites of magnetic CoFe_2_O_4_ and non-magnetic TiO_2_ have significantly less Ms (emu/g) values than the pure spinel ferrite: CoFe_2_O_4_ (87.6), TC5 (3.1), TC10 (8.4), and TC15 (14.2). Furthermore, it comes as no surprise that composites with a greater percentage of magnetic spinel content present higher Ms values. This decrease, however, does not impair the capability of the magnetic composites to be easily removed from a solution through simple magnetic means (e.g., a magnetic bar).

### 3.6. Photocatalytic Oxidation Activity

#### 3.6.1. Photocatalytic Degradation of ODP under UV Light

Under UV light irradiation, the photocatalytic oxidation performance of the produced materials was evaluated against six organic dyes (Figure 10). The TCX composites exhibit higher dye degradation efficiencies than either starting material in all cases. This efficiency is due to increased charge carrier separation at the TiO_2_/CoFe_2_O_4_ contact, which enhances the electron lifetime for TCX samples [42]. 

#### 3.6.2. Photocatalytic Degradation of TL under UV and Visible Light

The photocatalytic oxidation of TL over TiO_2_, CoFe_2_O_4_, TC5, TC10, and TC15 was evaluated under UV and visible light illumination (Figure 11a,b). It is obvious that the cobalt ferrite (CF) present in the TC nanocomposite maintains and slightly enhances the photocatalytic activity of titanium dioxide under both UV and visible light. The highest TL photocatalytic oxidation was observed for TC10, which was 75.31% at 180 min under UV illumination and 50.4% at 180 min under visible light illumination. Pure titania and cobalt ferrite, on the other hand, exhibit minor photocatalytic efficiency when compared to TiO_2_/CoFe_2_O_4_ composites. This means that cobalt ferrite is crucial in enhancing the photocatalytic oxidation process. The high photocatalytic oxidation of the TC composites, particularly the TC10 sample, can be attributed to the presence of ferrites nanoparticles, which can improve light absorption and ensure more efficient charge separation.

We performed recyclability experiments to verify the stability and reusability of the TC10 toward the TL photodegradation. Each time, the TC10 catalyst was collected and recovered, and then used in a cyclic batch. This procedure was conducted five times, with the final degradation efficiency recorded each time (Figure 11c). The TL degradation efficiency decreases slightly as the recycling test progresses, from ~75% in the first cycle to 65% in the final. As a result, we may consider this test to be indicative of the TC10 photocatalyst stability and reusability in the TL degradation.

Additionally, the synergistic interaction between TiO_2_ and CoFe_2_O_4_ in TC10 was the optimal compared to the rest of the composites as depicted in XRD, optical, vibrational, and TEM analysis. Trapping experiments with scavengers were carried out under UV light to elucidate the photocatalytic oxidation mechanism for the TL and to detect the reactive species (Figure 12). There was no change in the oxidation process when IPA (hydroxyl radical scavenger), and KBrO_3_ (electron scavenger) were added. In contrast, BQ (superoxide radical anion scavenger) had a noticeable effect, whereas the KI (hole scavenger) addition resulted in the photocatalytic oxidation reaction being blocked. Thus, holes were the prime motivator, followed by oxygen peroxide radicals as the main active species in the photocatalytic oxidation process (as observed in the graphical abstract).

The dispersion of ferrites over TiO_2_, keeping the mesoporous nature and good visible light absorption, in addition to the broad-spectrum range of the catalysts, which allowed them to absorb UV and solar radiation without the help of oxidizing agents, were further causes for the improved photocatalytic activities. 

For all dyes, significant degradation efficiency is achieved by the TCX samples: MB (over 95%), MO (45%), CV (over 83%), AO (over 91%), RhB (58%), Rh6G (56%), as will be shown in Figure 13. Among TCX samples, judging by overall performance, the best sample is TC10 (10% *w*/*w* CoFe_2_O_4_). It means that, the addition of CoFe_2_O_4_ has a volcano effect: it improves photocatalytic experiments up to an optimum (sample TC10), while a higher or lower amount decreases photocatalytic reactions.

## 4. Conclusions

In this study, hybrid nanocomposites comprised of CoFe_2_O_4_ loaded into TiO_2_ (5–15% *w*/*w*) were synthesized, characterized, and evaluated in the photocatalytic oxidation of six different organic dye pollutants under UV light and meanwhile under UV and visible light in the degradation of tetracycline. Characterization by XRD, FTIR, and Raman spectroscopy verified the fruitful construction of the CoFe_2_O_4_/TiO_2_ hybrid nanocomposites. The optical analysis revealed enhanced light absorption by the CoFe_2_O_4_/TiO_2_ nanocomposites compared to pure TiO_2_. In terms of oxidation/removal efficiencies under UV and visible light, CoFe_2_O_4_/TiO_2_ (10% *w*/*w*) outperformed the other tested hybrid nanocomposites in the tested six organic azo dyes and the antibiotic tetracycline in the absence of any oxidant. The main reason for the observed behavior was the improved photo-charge carrier separation at the interfaces of TiO_2_/CoFe_2_O_4_ nanocomposites, which is necessary for boosting photocatalytic activity. As long as the primary holes of the reactive species are present, the photocatalytic oxidation can be occurred. As a result, the CoFe_2_O_4_/TiO_2_ catalyst substantially increased the oxidative photocatalytic activity. Finally, the CoFe_2_O_4_/TiO_2_ powder is capable of easily being retrieved from the treated solution with a magnetic means as to its magnetic characteristics.

## Figures and Tables

**Figure 1 nanomaterials-12-03290-f001:**
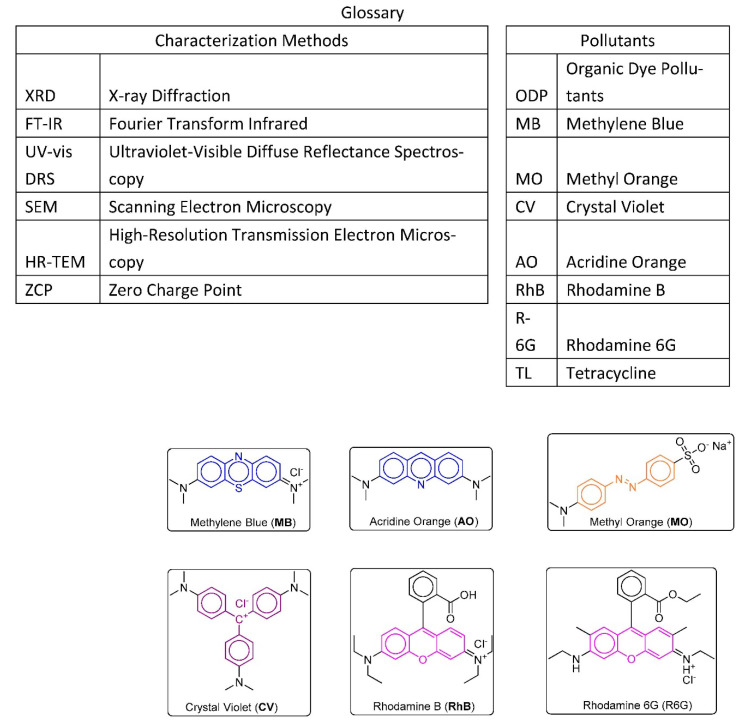
The structures of selected ODP.

**Figure 2 nanomaterials-12-03290-f002:**
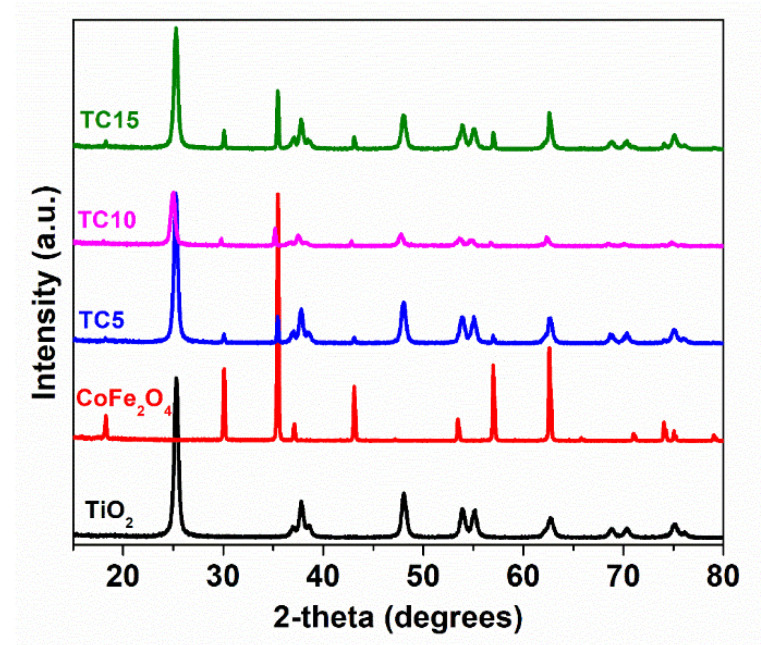
XRD patterns of the synthesized photocatalysts.

**Figure 3 nanomaterials-12-03290-f003:**
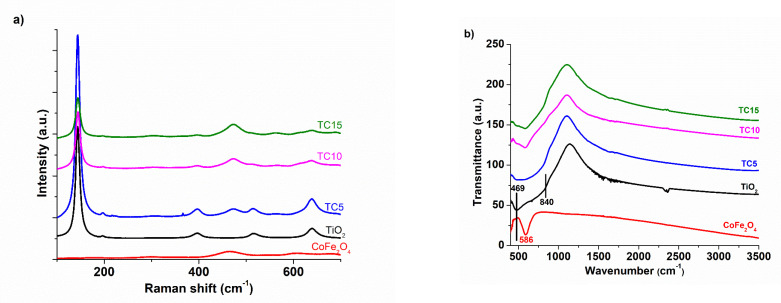
Vibrational spectrograms: (**a**) Micro-Raman and (**b**) FT-IR of synthesized photocatalysts.

**Figure 4 nanomaterials-12-03290-f004:**
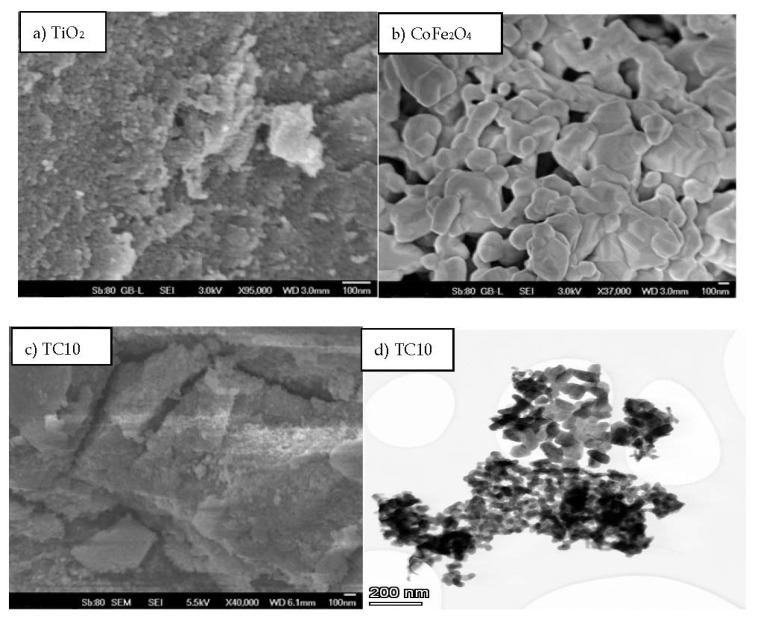
Microscopy-based analyses of the samples with SEM at 100 nm (**a**–**c**) and with TEM at 200 nm (**d**).

**Figure 5 nanomaterials-12-03290-f005:**
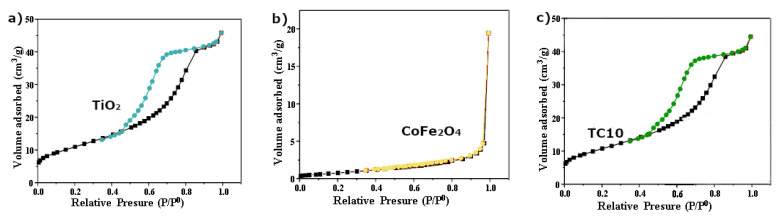
N_2_ adsorption-desorption isotherm plot of TiO_2_ (**a**), CoFe_2_O_4_ (**b**) and TC10 (**c**).

**Figure 6 nanomaterials-12-03290-f006:**
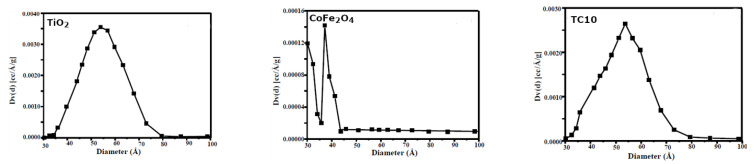
Pore size distribution curves of TiO_2_, CoFe_2_O_4_, and TC10.

**Figure 7 nanomaterials-12-03290-f007:**
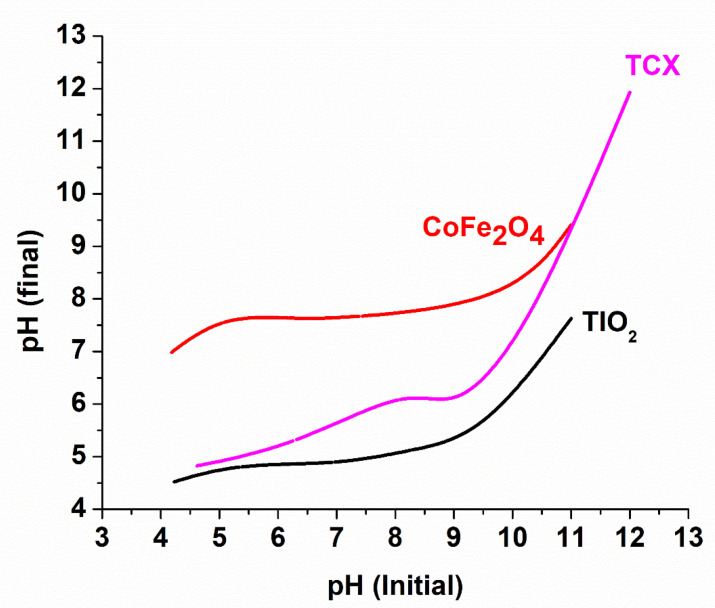
pHinitial vs. pHfinal curves for ZCP determination.

**Figure 8 nanomaterials-12-03290-f008:**
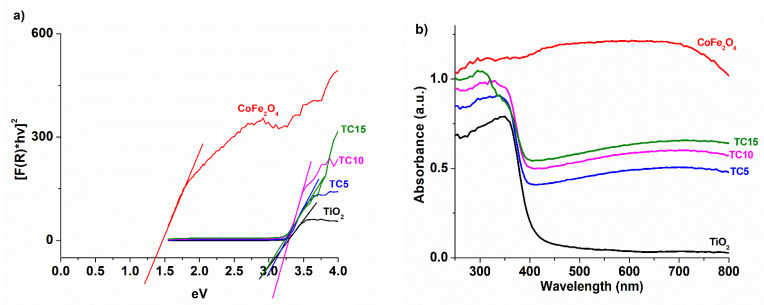
Tauc plots of the prepared photocatalysts (**a**), along with the relevant absorbance spectra (**b**).

**Figure 9 nanomaterials-12-03290-f009:**
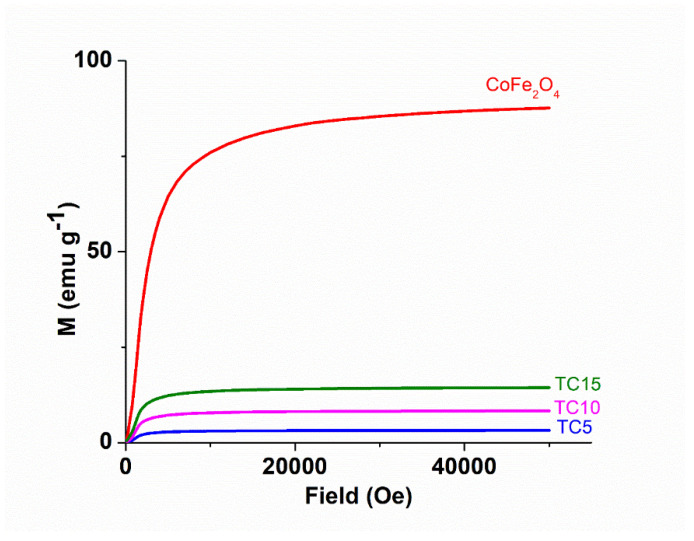
Magnetization curves of CoFe_2_O_4_ and the TCX composites.

**Figure 10 nanomaterials-12-03290-f010:**
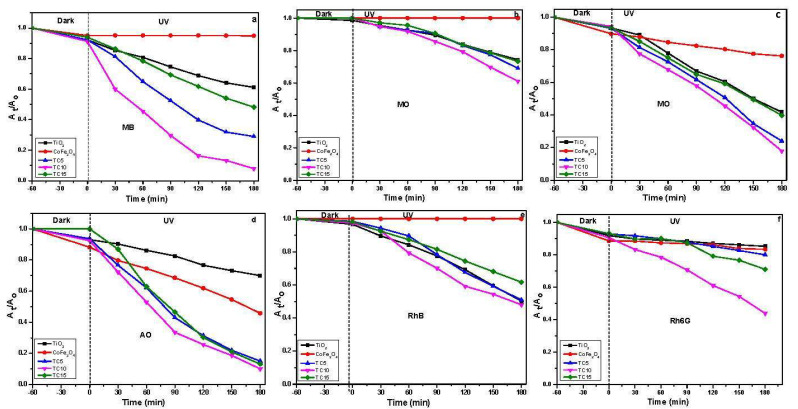
Photocatalytic oxidative degradation of (**a**) Methylene Blue (MB), (**b**) Methyl Orange (MO), (**c**) Crystal Violet (CV), (**d**) Acridine Orange (AO), and (**e**) Rhodamine B (RhB and (**f**)) Rhodamine 6G, under UV irradiation by the prepared samples.

**Figure 11 nanomaterials-12-03290-f011:**
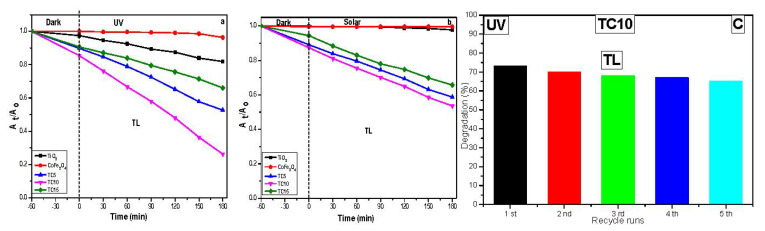
Photocatalytic oxidative degradation of tetracycline under (**a**) UV, (**b**) visible light irradiation using the prepared photocatalysts and (**c**) Reusability tests for the degradation efficiency of TL under UV for 5 subsequent cycles.

**Figure 12 nanomaterials-12-03290-f012:**
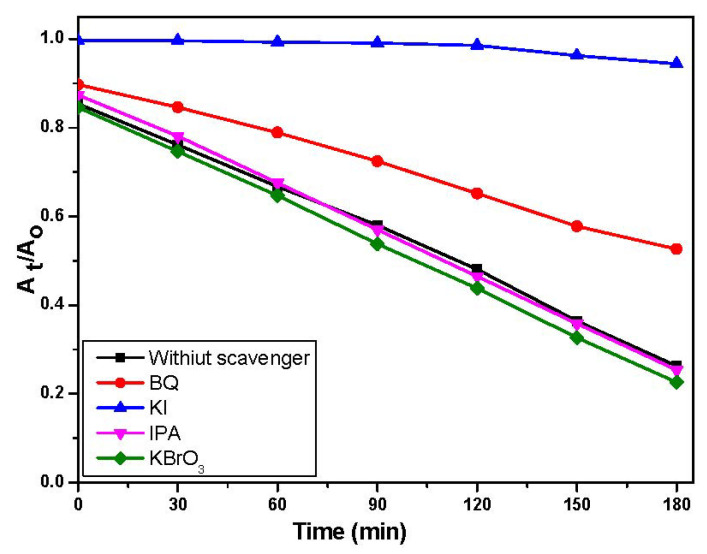
Scavengers’ effects on TL oxidation efficiency using the TC10 composite under UV irradiation.

**Figure 13 nanomaterials-12-03290-f013:**
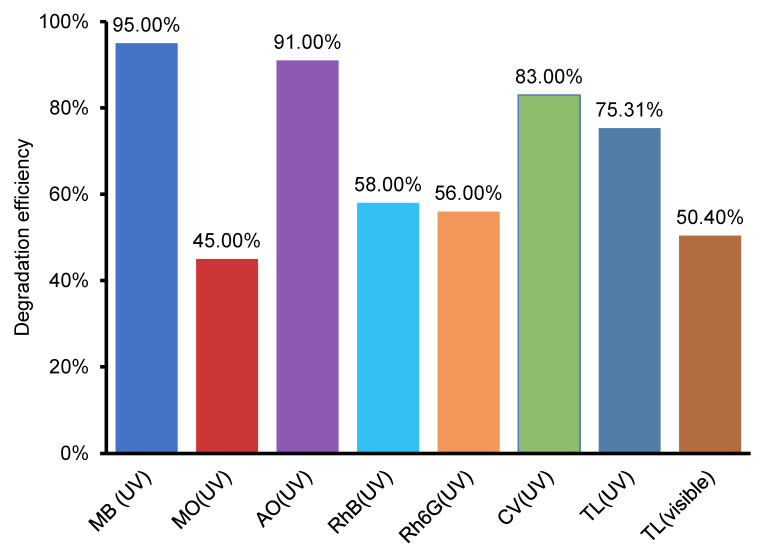
Significant degradation efficiency of all dyes and TL.

## Data Availability

Not applicable.

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
