# Peer review of "Magnetic TiO2/CoFe2O4 Photocatalysts for Degradation of Organic Dyes and Pharmaceuticals without Oxidants"

_nanomaterials, 2022, doi:10.3390/nano12193290_

Round 1
Reviewer 1 Report
The manuscript submitted by Ibrahim et al. describes the synthesis of magnetic TiO2/CoFe2O4 composite photocatalysts towards the degradation of various dye molecules and tetracycline under ultraviolet and visible light irradiation. The manuscript is not presented well, the introduction and the results are poorly written, which need significant modifications. Authors should carefully revise the manuscript based on the following comments before it can be considered further for publication.
1. In the first paragraph of the introduction section, authors have mentioned about the environmental issues of various organic dye pollutants. However, the issues related to pharmaceutical pollutants, which are a part of emerging pollutants (EPs) that are causing more menace is not described. Authors may refer and cite these papers https://doi.org/10.1016/j.surfin.2022.101941; https://doi.org/10.1016/j.cplett.2020.137435; https://doi.org/10.1016/j.materresbull.2021.111290 and https://doi.org/10.3390/catal11040426.
2. In line 64-65 of page 3, authors have written “Thus, the formation of composites with TiO2, where an additional component can rectify its aforementioned weaknesses, is a common strategy [1].” Authors should mention the importance of heterojunction formation that leads to enhanced charge separation and improved photocatalytic activity. Authors may refer and cite these papers https://doi.org/10.1016/j.apcatb.2013.05.077 and https://doi.org/10.3390/catal11040426 for rewriting this part.
3. The TEM micrograph presented in Fig. 4d corresponding to TC10 photocatalyst is of very poor quality and no scale is presented in the micrograph. Authors need to redo the TEM and also present HRTEM micrographs and index the fringes to find the d-spacing.
4. Authors should plot the isotherm data presented in Fig. 5 without clubbing such that the scale on the y-axis becomes visible to the readers. On the other hand, in line 175 of page 7, the authors have not mentioned the unit of the specific surface area of the synthesized photocatalysts.
5. Authors should not club the DRS spectrum and Tauc plot presented in Fig. 8. The DRS spectra and Tauc plots should be plotted as two separate graphs and can be named as (a) and (b) to be presented together as a single figure.
6. Authors have synthesized a magnetic photocatalyst that could be recycled. However, authors have not presented any data corresponding to the recyclability studies. Authors should refer to https://doi.org/10.1016/j.apsusc.2014.04.121, perform recyclability studies and present the data along with some photographs of the magnetic recyclability to make it attractive to the readers.
7. Authors have presented a plot in Fig. 13, which is not cited or explained in the manuscript.
Author Response
We are very grateful to the reviewer for his thoughtful comments and suggestions. We believe that our manuscript has been noticeably modified based on these revisions.
- In the first paragraph of the introduction section, authors have mentioned about the environmental issues of various organic dye pollutants. However, the issues related to pharmaceutical pollutants, which are a part of emerging pollutants (EPs) that are causing more menace is not described. Authors may refer and cite these papers https://doi.org/10.1016/j.surfin.2022.101941; https://doi.org/10.1016/j.cplett.2020.137435; https://doi.org/10.1016/j.materresbull.2021.111290 and https://doi.org/10.3390/catal11040426.
To further highlight the issues enveloping the pharmaceutical pollutants, the following sentence has been added [lines 50-51], along with a new reference from the suggested ones:
‘’Pharmaceutical ingredients are another major source of pollutants. They are considered a significant threat to humans' endocrine, reproductive, and cognitive systems [8].’’
- In line 67 of page 3, authors have written “Thus, the formation of composites with TiO2, where an additional component can rectify its aforementioned weaknesses, is a common strategy [1].” Authors should mention the importance of heterojunction formation that leads to enhanced charge separation and improved photocatalytic activity. Authors may refer and cite these papers https://doi.org/10.1016/j.apcatb.2013.05.077 and https://doi.org/10.3390/catal11040426 for rewriting this part.
Replay:
This part (lines 67-70) has been rewritten, to stress the role of heterojunctions with quoting suggested references [1,15 and16]:
‘’Thus, the formation of composites with TiO2, evolved into a common strategy since it can rectify its weaknesses [1]. The formation of such a photocatalytic heterojunction can enhance factors such as charge separation, thereby enhancing photocatalytic performance [15, 16].’’
- The TEM micrograph presented in Fig. 4d corresponding to TC10 photocatalyst is of very poor quality and no scale is presented in the micrograph. Authors need to redo the TEM and also present HRTEM micrographs and index the fringes to find the d-spacing.
Replay:
A new micrograph (4d) has been added to the revised manuscript denoting an enhancing quality.
Unfortunately, the indexed fringes are not available now to get d-spacing.
- Authors should plot the isotherm data presented in Fig. 5 without clubbing such that the scale on the y-axis becomes visible to the readers. On the other hand, in line 175 of page 7, the authors have not mentioned the unit of the specific surface area of the synthesized photocatalysts.
Replay:
Fig. 5 has been split into Fig 5 a-c to display their axes. Also, the units for the specific surface area have been added in line 183 as (m2/g).
|
|
Fig. 5. N2 adsorption-desorption isotherm plot of TiO2 (a), CoFe2O4 (bl, and TC10 (c).
- Authors should not club the DRS spectrum and Tauc plot presented in Fig. 8. The DRS spectra and Tauc plots should be plotted as two separate graphs and can be named as (a) and (b) to be presented together as a single figure.
Replay:
Fig. 8 has been split into Fig. 8a and b, based on the reviewer’s request.
Fig. 8. Tauc plots of the prepared photocatalysts (a), along with the relevant absorbance spectra (b).
- Authors have synthesized a magnetic photocatalyst that could be recycled. However, authors have not presented any data corresponding to the recyclability studies. Authors should refer to https://doi.org/10.1016/j.apsusc.2014.04.121, perform recyclability studies and present the data along with some photographs of the magnetic recyclability to make it attractive to the readers.
Replay:
The recyclability test has been done and added to the revised manuscript, lines 250-255, together with figure 11c and its caption.
“We performed recyclability experiments to verify the stability and reusability of the TC10 toward the TL photodegradation. Each time the TC10 catalyst was collected and recovered then used in a cyclic batch. This procedure was done five times, with the final degradation efficiency recorded each time (Fig. 11c). The TL degradation efficiency decreases slightly as the recycling test progresses, from ~75% in the first cycle to 65% in the final. As a result, we may consider this test to be indicative of the TC10 photocatalyst stability and reusability in the TL degradation.”
- Authors have presented a plot in Fig. 13, which is not cited or explained in the manuscript.
Replay:
In line 233, we corrected the figure number.

Reviewer 2 Report
In this paper the photocatalytic activity of CoFe2O4/TiO2 composite was examined against 6 different dyes. Several characterization techniques are used and the authors concluded that 5% w/w exhibited the higher photocatalytic activity than either pure TiO2 or CoFe2O4.
The authors use different units for similar amounts e.g. nm and Å or oC and oC.
Again there are several abbreviations and it would be better to make a list rather than to run through the text.
In Fig.4 TEM on 200nm seems less focus than the 100nm SEM; why?
BET areas do not have units.
Pore size distribution method is not indicated.
Is the amount adsorbed in STP?
There are several such issues that give the impression that the paper didn’t prepare carefully.
Author Response
We appreciate greatly the minute and instructive comments and suggestions by the reviewer. We have revised our work to incorporate the suggested modifications.
The authors use different units for similar amounts e.g. nm and Å or oC and oC.
Replay:
All units were reviewed and edited throughout the manuscript to ensure unit uniformity.
1- Again there are several abbreviations and it would be better to make a list rather than to run through the text.
Replay:
The following abbreviation list has been added to the revised manuscript after line 56:
|
Glossary |
||||
|
Characterization Methods |
Pollutants |
|||
|
XRD |
X-ray Diffraction |
ODP |
Organic Dye Pollutants |
|
|
FT-IR |
Fourier Transform Infrared |
MB |
Methylene Blue |
|
|
UV-vis DRS |
Ultraviolet-Visible Diffuse Reflectance Spectroscopy |
MO |
Methyl Orange |
|
|
SEM |
Scanning Electron Microscopy |
CV |
Crystal Violet |
|
|
HR-TEM |
High-Resolution Transmission Electron Microscopy |
AO |
Acridine Orange |
|
|
ZCP |
Zero Charge Point |
RhB |
Rhodamine B |
|
|
R-6G |
Rhodamine 6G |
|||
|
TL |
Tetracycline |
|||
2-In Fig.4 TEM on 200nm seems less focus than the 100nm SEM; why?
Replay:
It is difficult to compare the focus disparity of TEM and SEM at two different measuring scales. However, we added a new TEM micrograph of TC10 as in figure 4d. It seems a little bit foggy, but the nanoscale particles are evident in this sample.
3-BET areas do not have units.
Replay:
The units of the BET measurements have been added.
4-Pore size distribution method is not indicated.
Replay:
Pore size distribution method is inserted (lines 119-123); we added the following paragraph:
“Textural characterization of the samples was carried out by the N2 adsorption-desorption at −196 °C. Prior the measurements, the samples were outgassed at 200 °C for 48 h. The Brunauer–Emmett–Teller (BET) equation was applied to determine the specific surface area (SBET). The BJH method was applied to the desorption branch of the N2 isotherms to obtain the meso-pore volume (Vmeso) and cumulative surface area (SBJH) of the mesopores.”
5-Is the amount adsorbed in STP?
Replay:
“Yes, where the amount adsorbed in STP often refers to the pressure of the room and the temperature of the adsorption manifold.”

Reviewer 3 Report
The manuscript entitled “Magnetic TiO2/CoFe2O4 photocatalysts for degradation of organic dyes and pharmaceuticals without oxidants” reports the synthesis of TiO2/CoFe2O4 photocatalyst. In addition, a complete characterization of photocatalytic material and accurate photocatalytic degradation studies of six organic dye pollutants and tetracycline are described. The physical-chemical characterization of the synthetized materials has been well performed and described however, photocatalysis studies should be better presented.
In my opinion the manuscript is suitable for publication in Nanomaterials journal after a minor revision to increase the reader’s interest and the accuracy of the research.
The following concerns should be addressed:
1) In the introduction session, the novelty and significance of the work should be emphasised. In addition, the potential impact of the research and why it is important, compared to other research in this field or previous studies, should be discussed.
2) Please pay attention to some typos.
3) The names of the samples TC5, TC10, and TC15 should be better described. What is the meaning of the acronym TC?
4) Concentration and volume of the dyes and tetracycline solutions used for the photodegradation studies should be reported. Similarly, the amount of catalysts used for these experiments should also be specified.
5) Did the authors evaluate the effect of pollutants concentration and of photocatalyst dose on degradation processes? These aspects are important to improve the efficiency of the process.
6) To have very efficient materials for photodegradation processes with a good recyclability, catalysts should also be reused in subsequent photocatalytic cycles. Indeed, the recyclability of a catalyst is a very important aspect for the feasibility and cheapness of the material studied. The Authors believe that the same photocatalysts could be reused in different cycles to degrade the ODPs or TL?
Author Response
We would like to thank the reviewer for his thoughtful comments and efforts towards improving our manuscript. We believe that the manuscript has been noticeably modified based on these revisions.
1) In the introduction session, the novelty and significance of the work should be emphasised. In addition, the potential impact of the research and why it is important, compared to other research in this field or previous studies, should be discussed.
Replay:
The following sentences have been added (lines 76-79), giving a clear declaration about the significance of the work:
‘This work presents a complete overview of the TiO2/spinel ferrite nanocomposites with an extensive evaluation of their photocatalytic capabilities against several of the more well-known organic pollutants. It also presents the potential of the nanocomposites for recyclable usage, taking advantage of their magnetic retrieval ability, thereby highlighting the extensive capacities of the TiO2/spinel ferrite nanocomposites.’’
2) Please pay attention to some typos.
Replay:
The manuscript has been carefully revised.
3) The names of the samples TC5, TC10, and TC15 should be better described. What is the meaning of the acronym TC?
Replay:
In lines 110 and 111, the following sentence has been rectified to show the meaning of TC5-15:
“After washing and drying, we acquired the TC5, TC10, and TC15 samples, each with an index signifying its % w/w of CoFe2O4 (C) and TiO2 (T).”
4) Concentration and volume of the dyes and tetracycline solutions used for the photodegradation studies should be reported. Similarly, the amount of catalysts used for these experiments should also be specified.
Replay:
Lines 123-124 have been amended to offer a clear description of the photodegradation testing parameters:
“5 mg of the photocatalyst was added to an ODP or TL solution (50 mL, 10 ppm) to evaluate its photocatalytic degradation performance.”
5) Did the authors evaluate the effect of pollutants concentration and of photocatalyst dose on degradation processes? These aspects are important to improve the efficiency of the process.
Replay:
This work aimed to study the wide effectiveness of the TiO2/spinel ferrite nanocomposites in the photocatalytic degradation of seven pollutants. The optimal concentrations of either catalyst dose or pollutant concentration were tested to allow for a direct comparison of the performance of catalysts. Then, we applied such experimental conditions to give the maximum degradation efficiency.
6) To have very efficient materials for photodegradation processes with a good recyclability, catalysts should also be reused in subsequent photocatalytic cycles. Indeed, the recyclability of a catalyst is a very important aspect for the feasibility and cheapness of the material studied. The Authors believe that the same photocatalysts could be reused in different cycles to degrade the ODPs or TL?
Replay:
The recyclability test has been done and added to the revised manuscript, lines 250-255, figure 11c with its caption.
“We performed recyclability experiments to verify the stability and reusability of the TC10 toward the TL photodegradation. Each time the TC10 catalyst was collected and recovered then used in a cyclic batch. This procedure was done five times, with the final degradation efficiency recorded each time (Fig. 11c). The TL degradation efficiency decreases slightly as the recycling test progresses, from ~75% in the first cycle to 65% in the final. As a result, we may consider this test to be indicative of the TC10 photocatalyst stability and reusability in the TL degradation.”

Round 2
Reviewer 1 Report
The authors have revised the manuscript satisfactorily and therefore it may be considered for publication.